# A Random Forest Approach to Estimate Daily Particulate Matter, Nitrogen Dioxide, and Ozone at Fine Spatial Resolution in Sweden

**Massimo Stafoggia [1,2,\*], Christer Johansson [3,4], Paul Glantz [4], Matteo Renzi [1],
Alexandra Shtein [5], Kees de Hoogh [6,7], Itai Kloog [5], Marina Davoli [1], Paola Michelozzi [1] and
Tom Bellander [2,8]**

[1] Department of Epidemiology, Lazio Region Health Service/ASL Roma 1, Via Cristoforo Colombo 112,
00147 Rome, Italy; m.renzi@deplazio.it (M.R.); m.davoli@deplazio.it (M.D.); p.michelozzi@deplazio.it (P.M.)
[2] Institute of Environmental Medicine (IMM), Karolinska Institutet, Nobelsväg 13, 17177 Stockholm, Sweden;
tom.bellander@ki.se
[3] Environment and Health Administration, Fleminggatan 4, Box 8136, 104 20 Stockholm, Sweden;
Christer.Johansson@aces.su.se
[4] Department of Environmental Science, Stockholm University, Svante Arrhenius Väg 8, 106 91 Stockholm,
Sweden; paul.glantz@aces.su.se
[5] Department of Geography and Environmental Development, Ben-Gurion University of the Negev, P.O.B. 653
Beer Sheva, Israel; shtien@post.bgu.ac.il (A.S.); ikloog@bgu.ac.il (I.K.)
[6] Swiss Tropical and Public Health Institute, 4002 Basel, Switzerland; c.dehoogh@swisstph.ch
[7] University of Basel, 4001 Basel, Switzerland
[8] Center for Occupational and Environmental Medicine, Stockholm Region, Solnavägen 4, 113 65 Stockholm,
Sweden
\* Correspondence: m.stafoggia@deplazio.it; Tel.: +39-06-9972-2174

**Abstract:** Air pollution is one of the leading causes of mortality worldwide. An accurate assessment of its spatial and temporal distribution is mandatory to conduct epidemiological studies able to estimate long-term (e.g., annual) and short-term (e.g., daily) health effects. While spatiotemporal models for particulate matter (PM) have been developed in several countries, estimates of daily nitrogen dioxide ($NO_2$) and ozone ($O_3$) concentrations at high spatial resolution are lacking, and no such models have been developed in Sweden. We collected data on daily air pollutant concentrations from routine monitoring networks over the period 2005–2016 and matched them with satellite data, dispersion models, meteorological parameters, and land-use variables. We developed a machine-learning approach, the random forest (RF), to estimate daily concentrations of $PM_{10}$ (PM<10 microns), $PM_{2.5}$ (PM<2.5 microns), $PM_{2.5-10}$ (PM between 2.5 and 10 microns), $NO_2$, and $O_3$ for each squared kilometer of Sweden over the period 2005–2016. Our models were able to describe between 64% ($PM_{10}$) and 78% ($O_3$) of air pollutant variability in held-out observations, and between 37% ($NO_2$) and 61% ($O_3$) in held-out monitors, with no major differences across years and seasons and better performance in larger cities such as Stockholm. These estimates will allow to investigate air pollution effects across the whole of Sweden, including suburban and rural areas, previously neglected by epidemiological investigations.

**Keywords:** air pollution; epidemiology; machine learning; nitrogen dioxide; ozone; particulate matter; random forest

---

## 1. Introduction

Air pollution is a major risk factor to human health, causing >4 million premature deaths every year worldwide, with more than 90% of the population living in areas exceeding the guideline limits from the World Health Organization [1].

The health effects of air pollution have been extensively documented in the epidemiological literature, and they have been broadly distinguished into acute effects stemming from short-term (e.g., daily) exposures [2–4] and chronic effects induced by long-term (e.g., annual) exposures [5]. In the former, the hypothesis is that day-to-day variability in air pollutants is causally related to daily peaks in mortality (or morbidity) outcomes, whereas in the latter it is assumed that residing in areas with larger-than-average air pollution exposures will increase adverse health effects in the long run. It is therefore necessary to characterize air pollutant distributions over space and time in order to design proper epidemiological studies able to disentangle acute and chronic effects.

Most of the evidence on the health effects of air pollution has focused on particulate matter (PM), especially the fine fraction ($PM_{2.5}$), and previous studies have generally been conducted in urban areas due to lack of observations or reliable model estimates for suburban or rural areas [5–7]. This is a limitation, since many people live in non-urban areas characterized by a different source profile of air pollution compared to cities [8]. In addition, access to healthcare facilities can be more problematic in remote areas posing a greater risk to the most vulnerable and isolated individuals [9]. Finally, concentrations of $PM_{2.5}$ and nitrogen dioxide ($NO_2$) are expected to be lower away from the major cities, and most of recent research is trying to understand whether there exist health effects from air pollution that then require revision of the air quality standards.

$NO_2$ is a traffic-generated air pollutant that has been related to both acute and chronic effects on humans [10,11]. Most of the short-term studies have used crude estimates of daily exposures based on central monitoring stations, whereas long-term studies have defined exposures based on estimates from land-use regression, dispersion models, or hybrid approaches [12]. However, research on the health effects of $NO_2$ exposures in smaller cities or suburban and industrial regions is lacking. This is likely because reliable estimates of spatial and temporal variability of $NO_2$ concentrations over large geographical domains are few. While in principle there is no reason to believe that $NO_2$ effects, per unit change, should differ between urban and non-urban areas, in practice this may occur, because the composition of the underlying populations living in cities or out of them might be substantially different.

Tropospheric ozone is one of the most toxic components of the photochemical air pollution mixture. It is an oxidant air pollutant generated by photochemical reactions involving nitrogen oxides and volatile organic compounds. Short-term effects of ozone on mortality and morbidity have been reported in the epidemiological literature, among others from large multi-center studies conducted in Europe [13,14], the United States [15], and China [16]. The effects of long-term exposure to ozone on human health has however not been fully established [17]. Ozone levels are much higher today than in the pre-industrial era, and there are concerns of future increases related to global warming [18]. However, predicting ozone concentrations at fine spatial and temporal concentrations is extremely difficult because many parameters related to local sources, land-use characteristics, and meteorological conditions are involved in ozone formation and removal, resulting in high spatial and temporal variability [19].

We aimed to develop a new multi-stage methodology based on a machine-learning method—random forest (RF)—to estimate PM (10, 2.5, and 2.5–10), $NO_2$, and ozone ($O_3$) with high temporal (daily) and spatial (1-km$^2$) resolution across the whole of Sweden for the period 2005–2016. The method, already tested in Italy for PM [20], has for the first time characterized population exposure to multiple air pollutants also in areas with very low concentrations. The results obtained will allow investigators to study short-term and long-term effects of air pollution on human health at the national level in Sweden.

## 2. Data and Methodology

### 2.1. Study Region

Sweden belongs to northern Europe, located between the Baltic Sea (south and east), Finland (east), and Norway (north and west). With its approximately 450,000 square kilometers, it is the largest country in northern Europe and the 4th largest country of Europe. Sweden is characterized by a long coastal line and the presence of many lakes and rivers. Around 65% of Sweden's total land area is covered with forests. The highest population density is in southern Sweden, while the northern part encompasses almost 60% of the country area and is only sparsely populated. For the aims of this study, we defined a regular grid of 1-km$^2$ resolution over Sweden, for a total of 460,296 grid cells. In addition, in order to obtain finer estimates of daily air pollutants for Stockholm County, we nested a finer grid of cells sized $200 \times 200$ m in this area, for a total of 180,025 pixels.

### 2.2. Air Pollution Data

Data of daily air pollution concentrations were provided by the air quality database of the Swedish Meteorological and Hydrological Institute (SMHI). The urban data were from regulatory monitoring networks according to the requirements of the EU Air Quality Directive 2008/50/EC using reference instruments (or equivalent). Data from measurements located in rural areas were from the Co-operative Programme for Monitoring and Evaluation of the Long-range Transmission of Air Pollutants in Europe [21].

During 2005 to 2016, 180 monitoring sites in Sweden collected data on PM, 144 on $NO_2$, and 52 on $O_3$, with higher coverage in southern Sweden and in later years (67, 53, and 27 sites in 2016 and 53, 51, and 20 sites in 2005). The spatial distribution of the ground-based sites is presented in Figure 1, while the number of monitors and descriptive statistics per year and pollutant are reported in Table 1.

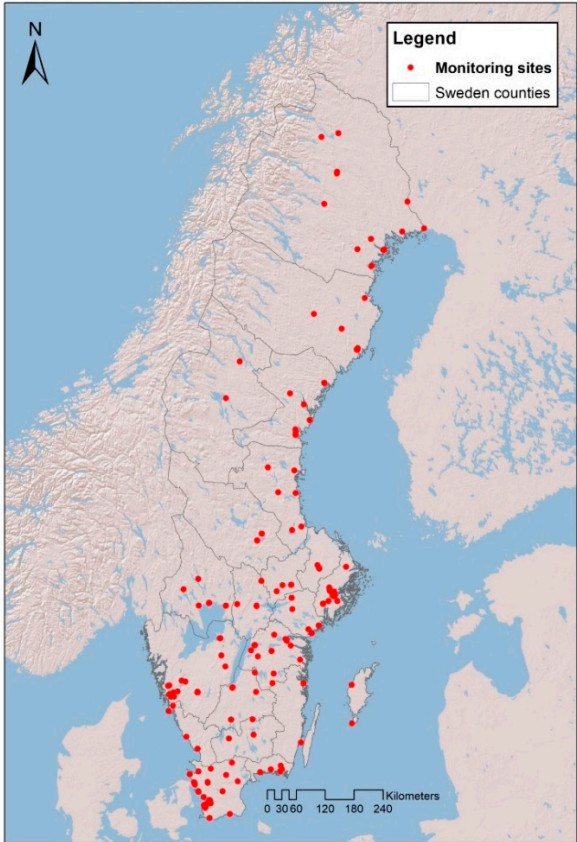

**Figure 1.** Spatial distribution of the monitoring stations in Sweden, years 2005–2016.

**Table 1.** Number of monitors per year and pollutant, and descriptive statistics.

| Year | $PM_{10}$ | | | $PM_{2.5}$ | | | $NO_2$ | | | $O_3$ | | |
|------|-----------|--------|--------------------------|------------|--------|--------------------------|--------|--------|--------------------------|--------|--------|--------------------------|
| | No. of Stations | Median | 25th–75th Percentiles | No. of Stations | Median | 25th–75th Percentiles | No. of Stations | Median | 25th–75th Percentiles | No. of Stations | Median | 25th–75th Percentiles |
| 2005 | 61 | 15.6 | 9.9–24.2 | 7 | 10.3 | 7.8–14.4 | 60 | 15.4 | 7.9–27.2 | 23 | 56.9 | 43.6–70.4 |
| 2006 | 72 | 16.8 | 11.1–25.4 | 17 | 10.5 | 7.4–15.1 | 67 | 17.2 | 8.7–29.9 | 29 | 58.9 | 45.4–71.5 |
| 2007 | 64 | 15.6 | 10.1–24.0 | 18 | 8.1 | 5.6–11.3 | 55 | 15.2 | 7.9–27.9 | 29 | 55.1 | 43.7–66.5 |
| 2008 | 58 | 15.3 | 9.7–23.1 | 17 | 7.9 | 5.3–11.3 | 60 | 16.3 | 8.4–28.3 | 24 | 54.6 | 41.3–68.0 |
| 2009 | 54 | 14.3 | 9.2–21.3 | 25 | 6.2 | 4.0–9.5 | 58 | 16.5 | 8.7–28.2 | 26 | 53.9 | 42.1–65.8 |
| 2010 | 61 | 13.4 | 8.6–20.2 | 24 | 6.0 | 3.8–9.5 | 58 | 19.1 | 8.8–33.2 | 26 | 55.6 | 43.0–66.9 |
| 2011 | 59 | 15.0 | 9.6–23.2 | 25 | 6.0 | 3.7–9.9 | 58 | 18.1 | 8.3–31.0 | 27 | 57.0 | 43.2–70.2 |
| 2012 | 60 | 12.7 | 8.4–19.4 | 24 | 5.0 | 3.1–8.1 | 60 | 18.1 | 9.0–30.0 | 22 | 51.7 | 39.3–64.8 |
| 2013 | 66 | 13.4 | 8.5–20.4 | 21 | 5.0 | 3.1–7.6 | 58 | 18.3 | 9.6–31.2 | 30 | 55.3 | 43.8–67.9 |
| 2014 | 63 | 13.7 | 8.7–20.8 | 28 | 5.8 | 3.6–9.1 | 50 | 17.2 | 8.8–28.9 | 30 | 54.2 | 42.0–65.2 |
| 2015 | 55 | 12.0 | 8.0–18.1 | 27 | 4.7 | 3.1–7.0 | 45 | 16.6 | 7.8–29.0 | 30 | 55.9 | 44.7–66.0 |
| 2016 | 62 | 11.4 | 7.4–17.6 | 29 | 4.5 | 2.8–7.1 | 53 | 17.5 | 8.6–29.5 | 30 | 52.4 | 40.7–63.6 |
| 2005–2016 | 172 | 13.9 | 8.9–21.3 | 59 | 6.0 | 3.7–9.5 | 141 | 17.1 | 8.5–29.6 | 45 | 55.1 | 42.7–67.2 |

### 2.3. Spatiotemporal Predictor Variables

We collected a number of spatiotemporal predictor variables aimed at capturing variability in air pollution concentrations due to complex interactions between spatial and temporal components. These are defined as variables (e.g., temperature) that vary on a daily basis and between grid cells.

*Aerosol Optical Depth (AOD).* AOD is a measure of optical aerosol loading (the amount of light absorbed or scattered by suspended particles) and is expected to be related to the number of aerosol particles, larger than 0.1 micron, in a column of air. NASA has recently developed an aerosol retrieval algorithm, the Multi-Angle Implementation of Atmospheric Correction (MAIAC), which provides AOD data at 1-km$^2$ spatial resolution [22,23]. Similar to the approach applied for Italy [20,24], here we used MAIAC AOD data derived from Collection 6 MODIS Aqua level 1 data for the period 2005–2016. Since MAIAC AOD data are not represented for many days and over many areas in Sweden, we also used modelled AOD data from the Monitoring Atmospheric Composition and Climate–Interim Implementation (MACC-II) project. This project was developed within the Copernicus Atmosphere Monitoring Service (CAMS) and is available from the European Centre for Medium-Range Weather Forecasts (ECMWF) website [25]. AOD at five different wavelengths (469, 550, 670, 865, and 1240 nm) for all days within the period 2005–2016, at 0.125° × 0.125° (approximately 10 × 10-km$^2$) spatial resolution, were investigated here.

*Meteorological data.* Meteorological parameters (air and dew point temperature, sea-level barometric pressure, total cloud coverage, surface wind speed and direction, snow albedo, and planetary boundary layer (PBL) height) were retrieved by the ERA-Interim reanalysis project [26]. Data at the spatial resolution of 0.125° × 0.125° corresponding to 0:00 and 12:00 UTC for each day in 2005–2016 were included in the study.

*Atmospheric composition data.* We retrieved parameters of global atmospheric composition from ERA-Interim (total column ozone, 2005–2016), MACC re-analysis (PM$_{2.5}$, PM$_{10}$, and total column nitrogen oxides, 2005–2012), and CAMS near-real time models (PM$_{2.5}$, PM$_{10}$, and total column nitrogen dioxides, 2013–2016). Each parameter was downloaded for the 8 three-hour windows from 0:00 to 21:00 each day in 2005–2016, at the maximum spatial resolution available (0.125° × 0.125°).

*Normalized Difference Vegetation Index (NDVI).* We collected monthly estimates of NDVI from the MODIS NDVI product (MOD13A3) at 1-km$^2$ spatial resolution.

### 2.4. Spatial Predictor Variables

Spatial predictor variables are aimed at capturing variability in air pollution concentrations due to sources assumed constant over time (e.g., roads network).

Resident population. Data on the Swedish resident population for the year 2016 were provided by Statistics Sweden (SCB) for each of the 5985 demographic statistical areas (DeSO).

Imperviousness surface area (ISA). ISA is an indicator of the spatial distribution of artificial areas. For example, ISA includes housing areas, traffic areas (airports, harbors, railway yards, parking lots), roads, industrial and commercial areas, construction sites, etc. These data, with a spatial resolution of ~20 m and corresponding to year 2012, were downloaded from the Copernicus Land Monitoring Service (CLMS).

Light at night (LAN). LAN data are a proxy indicator for major conurbations and human activities. They were collected from the Visible Infrared Imaging Radiometer Suite (VIIRS) Day/Night Band (DNB), year 2015 [27], at a spatial resolution of ~750 m.

Land cover data. Land cover data were based on the Corine Land Cover (CLC) database of the year 2012 [28], and defined as percentage of each grid cell covered by eleven CLC classes (high/low development, urban green, industries, arable land, pastures, deciduous/evergreen/forest/shrubs, water).

Road density. Aggregated road density data at 1 km spatial resolution for "all" and "major" roads. The road data were originally obtained from the EuroStreets digital road network (version 3.1, based on TeleAtlas MultiNet TM for year 2008) and more details can be found in de Hoogh et al. [29] and Vienneau et al. [30].

Elevation. Mean elevation was downloaded from the European Digital Elevation Model (EU-DEM) provided by CLMS at 30 m spatial resolution.

## 2.5. Statistical Models

We developed a three-stage statistical methodology, based on random forests, as described in more detail by Stafoggia et al. [20]. Briefly, random forests are a family of machine-learning methods that consist in building an ensemble (or forest) of decision trees [31]. At any iteration, each tree is built using a bootstrap sample of the data, and each node of the tree is split according to a subset of randomly chosen predictors [32]. Finally, an optimal prediction of the target variable is obtained by averaging the outputs from each tree. The model also provides an estimate of the relative "importance" of each predictor, that is, how much the prediction squared error over all trees decreases after a variable is selected in the tree building process. We applied a different regression random forest model for each pollutant and stage of the analysis, as described below.

The first stage, only applied for PM, was aimed at establishing statistical relationships between daily $PM_{2.5}$ and $PM_{2.5-10}$ concentrations with co-located $PM_{10}$ measurements, in order to estimate fine and coarse PM at monitor sites and for days with data available only for $PM_{10}$. The outcome from this model was to produce an enlarged dataset for $PM_{2.5}$ and $PM_{2.5-10}$, to be used in stage 3. This was achieved by training the following regression random forest model:

$$PM_x \sim RF\ (PM_{10} + site\_location + month + day\_of\_week + latitude + longitude)$$

where $PM_x$ ($x$ being either "2.5" or "2.5–10") was related to co-located $PM_{10}$, location of the monitoring site (classified as either urban traffic, urban background, or rural), month, day of the week, and coordinates of the site. We added month and day of the week to capture residual temporal variations in air pollution due to seasonal and weekly patterns.

The second stage establishes a statistical relationship between observed MAIAC AOD and co-located modelled CAMS AOD, plus additional spatial and temporal predictors. The aim was to impute AOD in grid cells and for days with no MAIAC retrievals available, so a full spatiotemporal surface of AOD could be used in stage 3. The regression random forest model was the following:

$$MAIAC.AOD\ \sim\ RF\left(\sum_{k=1}^{5}\sum_{h=1}^{8}CAMS.AOD_{k,h}\ +\ day\_of\_year\ +\ latitude\ +\ longitude\right)$$

where MAIAC AOD was related to CAMS AOD at different bands $k$ and three-hour windows $h$, day of the year (from 1 to 365), and coordinates of the cell centroid. This stage is only relevant for PM modelling, as AOD was not used as a predictor for $NO_2$ or $O_3$ models.

Finally, the third stage aimed to establish relationships between daily air pollutant concentrations and AOD (for PM only), meteorology, atmospheric composition data, land use, and other predictors in order to estimate fields of PM, $NO_2$, and $O_3$ concentrations over areas where no monitoring stations were located. In addition, in order to account for autocorrelation of air pollutants over time (air pollution corresponding to present day being correlated with air pollution from previous day or days), we added lagged terms corresponding to three previous days for meteorological variables, air composition parameters, and AOD. We developed separate models for each pollutant over the whole period 2005–2016, as described below:

$$AirPollutant_{i,j}\ \sim\ RF\left(\sum_m X1_{i(j,\ j-1,\ j-2,j-3)}\ +\ \sum_n X2_i\right)$$

where the concentration of each air pollutant ($PM_{10}$, $PM_{2.5}$, $PM_{2.5-10}$, $NO_2$, or $O_3$) measured in grid cell $i$ on day $j$ was trained against spatiotemporal parameters (indexed by $m$) for the same cell and up to day $j$-3 (AOD (for PM models only), pollutant-specific atmospheric composition variable, meteorological parameters, planetary boundary layer height, NDVI, and spatial parameters (indexed by $n$) for the

same grid cell (resident population, ISA, LAN, CLC variables, elevation, length of all roads, length of major roads).

We checked the performance of each random forest model via "cross-validation" following two different approaches. First, we compared the predictions and observations from the "out-of-bag" (OOB) data of the random forest. In particular, each RF bootstrap dataset samples, on average, two thirds of the observations which are used to train the model ("in-bag" sample). The remaining third, called "out-of-bag" (OOB), is used as an external dataset for model validation. Second, since our objective was to estimate air pollutants over places with no monitoring stations, we also performed a cross-validation of the monitoring sites, that is, by randomly splitting the total set of monitors into ten groups. The model was applied on nine groups ("training" set) and predicted to the tenth group ("testing" set). We reiterated the procedure over the ten groups and finally checked the correlation between observed air pollutant concentrations and predictions in held-out monitors. The comparisons of both approaches were summarized in terms of $R^2$ (% of explained variance), root mean square error (RMSE), as well as intercept and slope (as measures of bias, obtained from a univariate linear regression between observations and cross-validated predictions) [20].

All statistical analyses were performed in R Version 3.6.0 (R Foundation for Statistical Computing, Vienna, Austria) using the package "ranger" for random forest models. GIS predictor variables were calculated using ArcGIS 10.5 (ESRI 2011. ArcGIS Desktop: Release 10. Redlands, CA, USA).

## 3. Results and Discussion

### 3.1. Monitored Data

The numbers of monitoring sites available in Sweden for each pollutant and year are reported in Table 1. These were the same for $PM_{10}$, $NO_2$, and $O_3$ during the study period, whereas numbers of sites measuring $PM_{2.5}$ concentrations have substantially increased over time, from 7 in 2005 to 29 in 2016. Across the whole period, $PM_{10}$ was measured in 98 sites located in proximity to traffic sources, 67 sites representing urban background concentrations, and 7 sites located in rural or remote areas. Corresponding numbers of sites were 26, 27, and 6 for $PM_{2.5}$, 69, 62, and 10 for $NO_2$, and 7, 19, and 19 for $O_3$ (data not shown).

Mean concentrations of $PM_{10}$ and $PM_{2.5}$ were very small and decreasing over time, whereas gas concentrations did not show any temporal trends.

### 3.2. Stages 1 and 2

The results of the stage 1 models predicting monitor-specific $PM_{2.5}$ and $PM_{2.5-10}$ concentrations from co-located $PM_{10}$ data for the period 2005–2016 are reported in Table S1 (Supplementary Materials). The linear correlations (as measured by Pearson's $\rho$ coefficient) with co-located $PM_{10}$ data were higher for coarse particles (ranging between $\rho = 0.82$ (2011) and $\rho = 0.93$ (2013)) than for fine particles (between $\rho = 0.52$ (2012) and $\rho = 0.71$ (2007)). As a consequence, stage 1 prediction models displayed a better performance for the coarse fraction, as reflected by the higher CV $R^2$, both in the OOB samples and in the left-out monitors.

Table S2 (Supplementary Materials) reports similar results for the stage 2 models, aimed at filling in missing data of MAIAC AOD using the co-located AOD estimates from CAMS as the main predictors. As displayed in the table, there were large missing fractions of MAIAC AOD data in Sweden. The relatively high linear correlations between co-located MAIAC and CAMS, in the order of $\rho = 0.7$, resulted in very good and stable stage 2 prediction models, with OOB CV $R^2$ ranging between 0.82 in 2016 and 0.88 in 2006, with negligible mean errors.

### 3.3. PM Results

CAMS predictions of $PM_{10}$, $PM_{2.5}$, and $PM_{2.5-10}$ were positively correlated with co-located measured concentrations (Table 2). The CAMS atmospheric composition variables were also among

the most important predictors in the stage 3 training models, possibly because they were able to predict both spatial and temporal variability of PM. As expected, planetary boundary layer showed a negative correlation with all particle metrics (the lower the mixing layer, the higher the ground-level concentrations), while barometric pressure was positively correlated with the particles (as higher pressure reflects stable conditions with little air circulation and consequent accumulation of pollutants from local sources). The north–south wind direction was only important in the training model for $PM_{2.5}$, while cloud coverage was negatively correlated with, and highly important in models for, $PM_{10}$ and $PM_{2.5-10}$. AOD was weakly correlated with all PM metrics and marginally important in the training models. Among the spatial predictors, proxies for urban areas (such as resident population, ISA, light at night, % urban area, road density) were positively correlated with PM, whereas variables describing natural land cover showed a negative correlation and a very limited importance in the training models. Interestingly, $PM_{2.5}$ concentrations were mostly explained by spatiotemporal covariates describing daily meteorological patterns, whereas $PM_{2.5-10}$ concentrations were better captured by spatial covariates representing urban settings (such as population density and impervious surfaces), and $PM_{10}$ data by a mix of both spatial and spatiotemporal variables (Table 2). Most of the aforementioned correlations, even when small in absolute values, where statistically significant ($p$-value < 0.05) because of the large number of observations analyzed.

The relationship between PM measurements and stage 3 predictions in OOB samples and left-out monitors are displayed in Figure 2a–c, and in Tables S3 and S4 (Supplementary Materials). In general, all the models provided unbiased predictions of PM, in both OOB samples and left-out monitors, resulting in univariate regression lines between observed and predicted PM with slopes close to one and intercepts close to zero (Figure 2). Model fit was better for $PM_{2.5}$ (CV-$R^2$ = 0.69 in OOB sample, 0.59 in left-out monitors), compared to $PM_{10}$ (0.64 and 0.50) and $PM_{2.5-10}$ (0.65 and 0.45). As expected, predictions in OOB samples captured higher percentages of PM variability, and introduced smaller errors, than the corresponding ones in left-out monitors. This is because, in the first approach, all monitors contributed with daily data in both training and testing datasets, whereas, in the second approach, separate monitors contributed the training and testing sets. Model fit statistics in both OOB samples (Table S3) and left-out monitors (Table S4) showed no major differences by year, season, and location of the monitors (urban traffic, urban background, rural), with good performance in the larger urban areas, such as Stockholm and Malmö.

Annual mean concentrations estimated for the year 2016 are displayed in Figure 3, and daily time series for the same year are shown in Figure 4. The $PM_{10}$ and $PM_{2.5}$ fields in Figure 3 show clear geographical variation, with increasing north–south and west–east gradients (with the largest urban areas being in southern Sweden and near the coast and the northwestern areas being characterized by large forests, mountain ranges, and remote isolated villages). $PM_{2.5-10}$ concentrations are highest near the coast and in the major cities, with no clear geographical variations with respect to north–south and west–east directions. Results for the other years investigated here are similar (not shown).

The time series displayed in Figure 4 show that the model (orange line) is capturing daily PM concentrations very well, as reflected in the comparisons with the measurements (blue line). The green line represents daily mean concentrations estimated for the whole of Sweden, which are lower than the values observed at the monitors, as these are usually located in populated areas characterized by higher-than-average concentrations.

**Table 2.** Results of the stage 3 model: Spearman's correlations between air pollutants and predictors, and relative importance (rank) of individual predictors in the random forest (RF) model. AOD, Aerosol Optical Depth; PBL, Planetary Boundary Layer; U, u component of the wind (horizontal wind toward east); V, v-component of the wind (horizontal wind towards north); NDVI, Normalized Difference Vegetation Index; ISA, Imperviousness Surface Areas; LAN, Light At Night.

| Predictor | $PM_{10}$ | | $PM_{2.5}$ | | $PM_{2.5-10}$ | | $NO_2$ | | $O_3$ | |
|---|---|---|---|---|---|---|---|---|---|---|
| | ρ | Importance (Rank) | ρ | Importance (Rank) | ρ | Importance (Rank) | ρ | Importance (Rank) | ρ | Importance (Rank) |
| **Spatiotemporal** | | | | | | | | | | |
| AOD | 0.05 | 14 | 0.13 | 15 | −0.01 | 13 | −0.05 | - | 0.15 | - |
| atmospheric composition var. | 0.35 | 1 | 0.44 | 1 | 0.21 | 4 | 0.12 | 12 | 0.35 | 3 |
| PBL (at midnight) | −0.14 | 8 | −0.14 | 13 | −0.10 | 9 | −0.21 | 6 | 0.09 | 2 |
| PBL (at midday) | 0.06 | 11 | −0.08 | 4 | 0.14 | 10 | −0.13 | 4 | 0.35 | 1 |
| wind U component | −0.02 | 15 | −0.09 | 7 | 0.03 | 15 | −0.02 | 7 | 0.05 | 5 |
| wind V component | 0.09 | 9 | 0.16 | 2 | 0.03 | 14 | 0.00 | 8 | −0.01 | 7 |
| air temperature | 0.02 | 17 | −0.01 | 14 | 0.04 | 17 | −0.13 | 16 | 0.12 | 4 |
| dew point temperature | −0.04 | 16 | −0.01 | 11 | −0.06 | 11 | −0.13 | 13 | −0.02 | 10 |
| cloud coverage | −0.17 | 3 | −0.04 | 9 | −0.20 | 2 | −0.06 | 18 | −0.21 | 13 |
| barometric pressure | 0.18 | 4 | 0.18 | 3 | 0.14 | 7 | 0.10 | 20 | −0.02 | 16 |
| snow albedo | 0.00 | 19 | 0.01 | 18 | −0.02 | 16 | −0.11 | - | −0.06 | - |
| NDVI | −0.13 | 10 | -0.11 | 8 | −0.12 | 5 | −0.31 | 15 | 0.07 | 11 |
| **Spatial** | | | | | | | | | | |
| resident population | 0.17 | 5 | −0.01 | - | 0.24 | 1 | 0.34 | 3 | −0.15 | 12 |
| ISA | 0.17 | 2 | 0.16 | 6 | 0.14 | 3 | 0.27 | 5 | −0.16 | - |
| LAN | 0.08 | 13 | −0.02 | 12 | 0.13 | 8 | 0.27 | 1 | −0.11 | 14 |
| elevation | −0.18 | 7 | −0.16 | 5 | −0.15 | 12 | −0.23 | 9 | 0.14 | 8 |
| all roads length | 0.17 | 6 | 0.10 | 10 | 0.18 | 6 | 0.44 | 2 | −0.16 | 15 |
| major roads length | 0.04 | - | 0.03 | - | 0.04 | - | 0.17 | 14 | −0.07 | - |
| % arable land | −0.05 | - | 0.01 | - | −0.07 | - | −0.14 | - | 0.01 | - |
| % deciduous | −0.04 | - | 0.01 | - | −0.07 | - | −0.18 | - | 0.05 | - |
| % evergreen | −0.17 | - | −0.12 | - | −0.16 | 21 | −0.29 | - | 0.15 | - |
| % forest | −0.09 | - | −0.08 | - | −0.08 | - | −0.17 | - | 0.06 | - |
| % industry | 0.02 | - | 0.02 | 17 | 0.01 | 19 | −0.03 | 17 | −0.01 | - |
| % pasture | 0.04 | - | 0.04 | - | 0.03 | - | −0.15 | - | 0.04 | - |
| % shrub | −0.12 | - | −0.11 | - | −0.09 | - | −0.19 | - | 0.05 | - |
| % urban area | 0.12 | 18 | 0.07 | 16 | 0.13 | 20 | 0.32 | 11 | −0.18 | 6 |
| % urban green | −0.10 | - | −0.09 | - | −0.09 | 18 | -0.15 | 19 | −0.03 | - |
| % water | 0.08 | 20 | 0.00 | - | 0.13 | 22 | 0.18 | 10 | −0.13 | 9 |

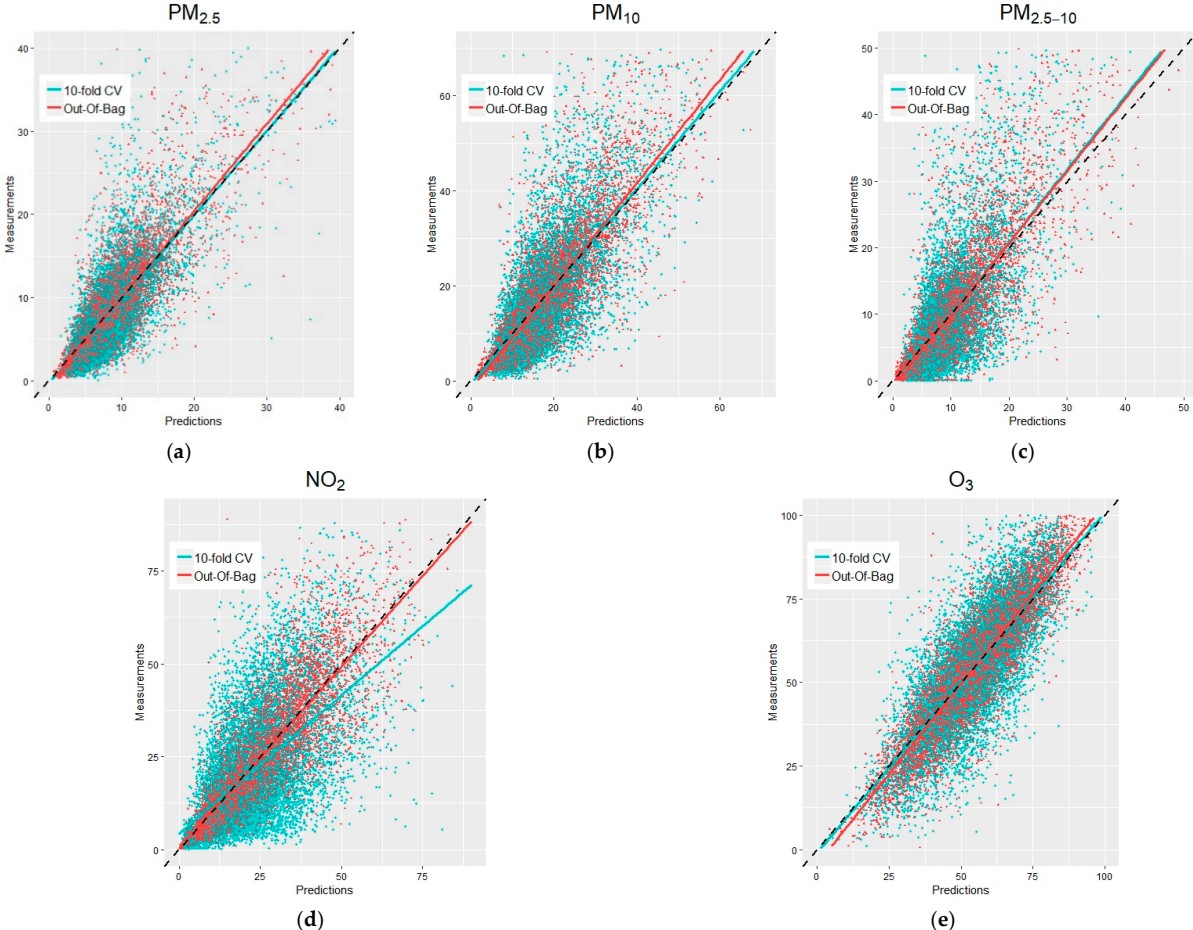

**Figure 2.** Results of the stage 3 model* for the five pollutants, PM$_{2.5}$ (**a**), PM$_{10}$ (**b**), PM$_{2.5-10}$ (**c**), NO$_2$ (**d**), and O$_3$ (**e**): measurements and predictions from "out-of-bag" (OOB) samples and 10-fold cross-validation by monitors. * Measurements (*y*-axis) vs. predictions (*x*-axis). The red and light blue lines represent the univariate regression lines between measurements and predictions in OOB samples or left-out monitors, respectively. Measurements are displayed on the *y*-axis as the purpose of the plot is to show how much variability in observations is captured, and how much bias introduced, by predictions.

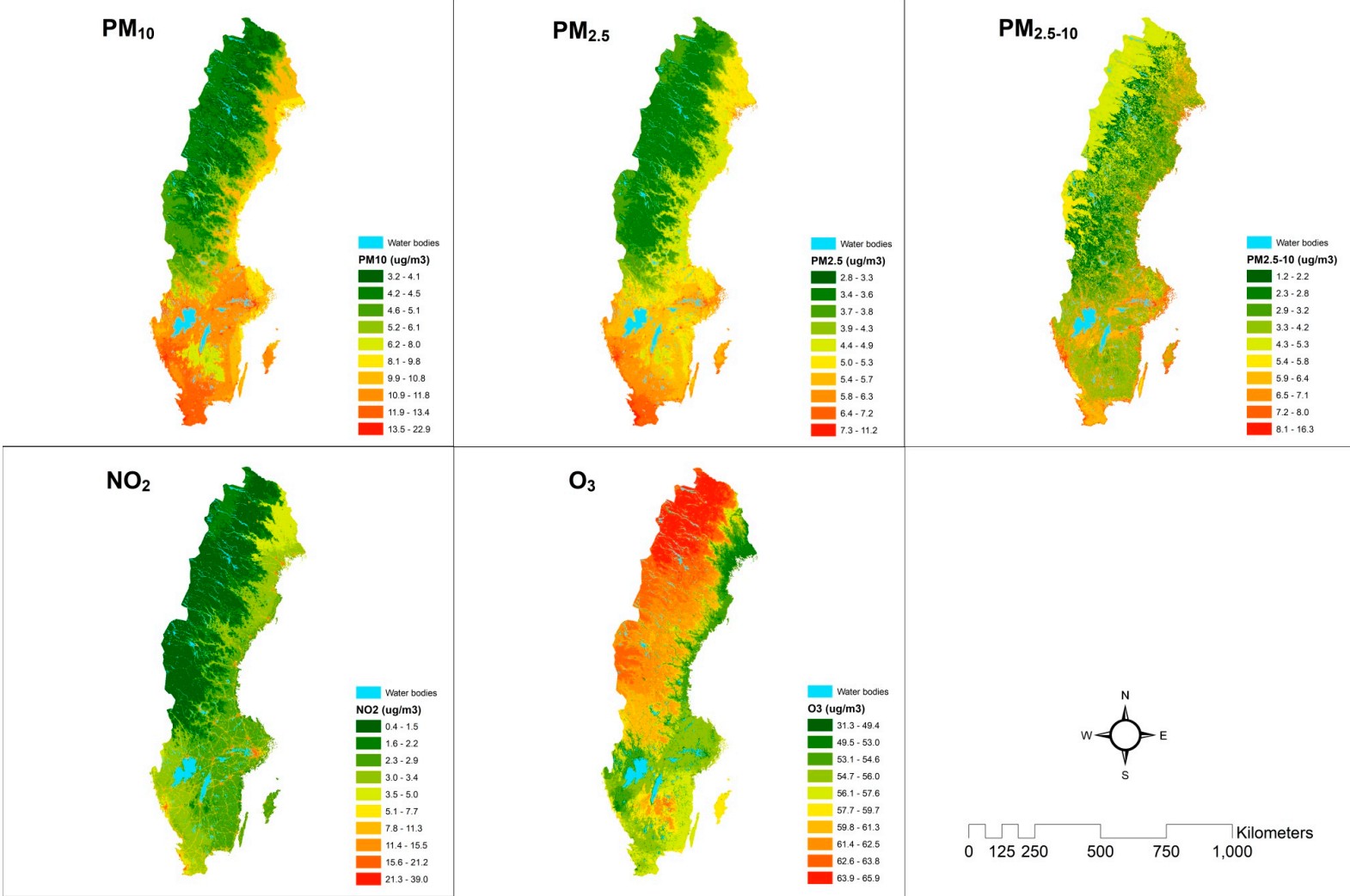

**Figure 3.** Fields of annual average air pollutant concentrations estimated for year 2016.

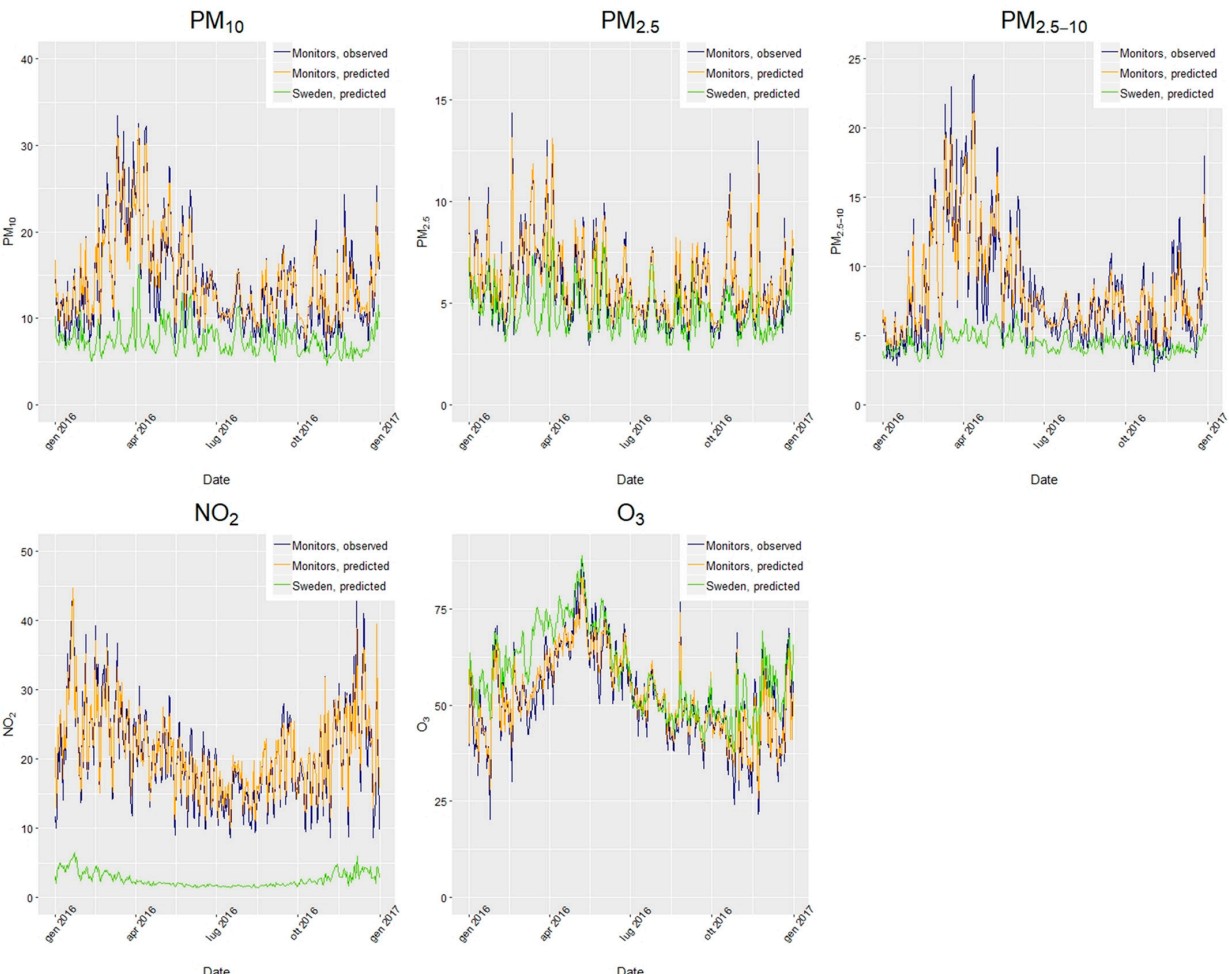

**Figure 4.** Time series of air pollutant concentrations: daily mean observations (blue line), daily mean predictions at the monitors (orange line), and daily mean predictions for the whole of Sweden (green line), year 2016.

### 3.4. NO₂ Results

Variability in $NO_2$ concentrations was mainly explained by spatial variables representing local sources in urban areas, such as LAN, road density, resident population, and ISA. Among the spatiotemporal predictors, Table 2 shows that PBL and wind components were to a high degree correlated with $NO_2$ (and important in the training models), whereas columnar $NO_2$ estimates from CAMS were marginally correlated with measurements and irrelevant in the training model. A consequence of the role played by spatial predictors is that the performance of the model differed in OOB samples or left-out monitors. In the first case a high percent of $NO_2$ variability was explained by the model ($R^2 = 0.74$), while in the second the $R^2$ decreased to 0.37, showing a limited ability of the model to predict full time series of $NO_2$ concentrations in external points (Table S4). This is apparent in Figure 2d, where the univariate regression lines relating $NO_2$ measurements with 10-fold CV (blue line) or OOB (red line) predictions deviate substantially, particularly for the former (blue points). This resulted in larger prediction errors for 10-fold CV estimates (12.9 $\mu g/m^3$ on average, Table S4) than for OOB estimates (8.3 $\mu g/m^3$, Table S3).

The mean predictions for 2016 are highest in the main cities and along the most important roadways (Figure 3, bottom left). This is also reflected in the daily time series for 2016, where the predicted and measured concentrations for the monitoring stations are much higher (around 25 $\mu g/m^3$) than those estimated for the whole of Sweden (around 3 $\mu g/m^3$).

### 3.5. O₃ Results

Ozone concentrations were highly positively correlated with spatiotemporal covariates such as PBL, total column $O_3$ from ERA-Interim, and daily mean temperature, whereas spatial covariates were only marginally correlated with $O_3$ observations and played a minor role in the training models. As for PM, there were mild differences in model fitting when considering OOB samples or left-out monitors, with little bias as apparent from Figure 2e, where both univariate regression lines do not depart from the 1:1 dashed line, and small mean prediction errors are observed (8.7 $\mu g/m^3$ for OOB estimates, Table S3, and 11.6 $\mu g/m^3$ for 10-fold CV estimates, Table S4).

The map and the time series for 2016 show, as expected, opposite trends compared with $PM_{10}$, $PM_{2.5}$, and $NO_2$, with highest estimated concentrations in remote and unpopulated areas, smallest concentrations along the coast and in the major cities (where the high concentrations of primary pollutants preclude the formation of ozone), and an inversed seasonality with spring–summer peaks and winter drops.

### 3.6. Comparison with Local Dispersion Models in Stockholm

Figure 5 and Table S5 (Supplementary Materials) present comparisons of concentrations of $PM_{10}$ and $NO_2$ predicted over Stockholm County by the stage 3 random forest and local dispersion models for the year 2015.

A description of the emission data, wind, and dispersion model used for the local modelling is provided in Segersson et al. [33], and it has been used to quantify exposure in several epidemiological and health impact assessment studies [34–38].

The two models predicted different geographical distributions of $PM_{10}$ concentrations, with higher levels in the archipelago and the western part of the county from the random forest, whereas the dispersion model predicted higher levels only around major roadways and in highly inhabited areas (Figure 5). Therefore, while the average predicted concentrations are similar between the two models (Table S5), the difference in the spatial distribution is relatively large, especially for the highest percentiles (Table S5). The correlation between the two predictions is weak ($\rho = 0.22$), and we have no clear explanation for that. Figure 5 shows that the $NO_2$ fields estimated for Stockholm County are, on the other hand, more similar between the two models ($\rho = 0.75$). This is likely because the main drivers of the $NO_2$ concentration and variability are associated with spatial terms (i.e., major roads and

combustion sources), equally captured by the statistical (RF) and deterministic (DM) approaches. This resulted in slightly higher concentrations estimated by the dispersion model.

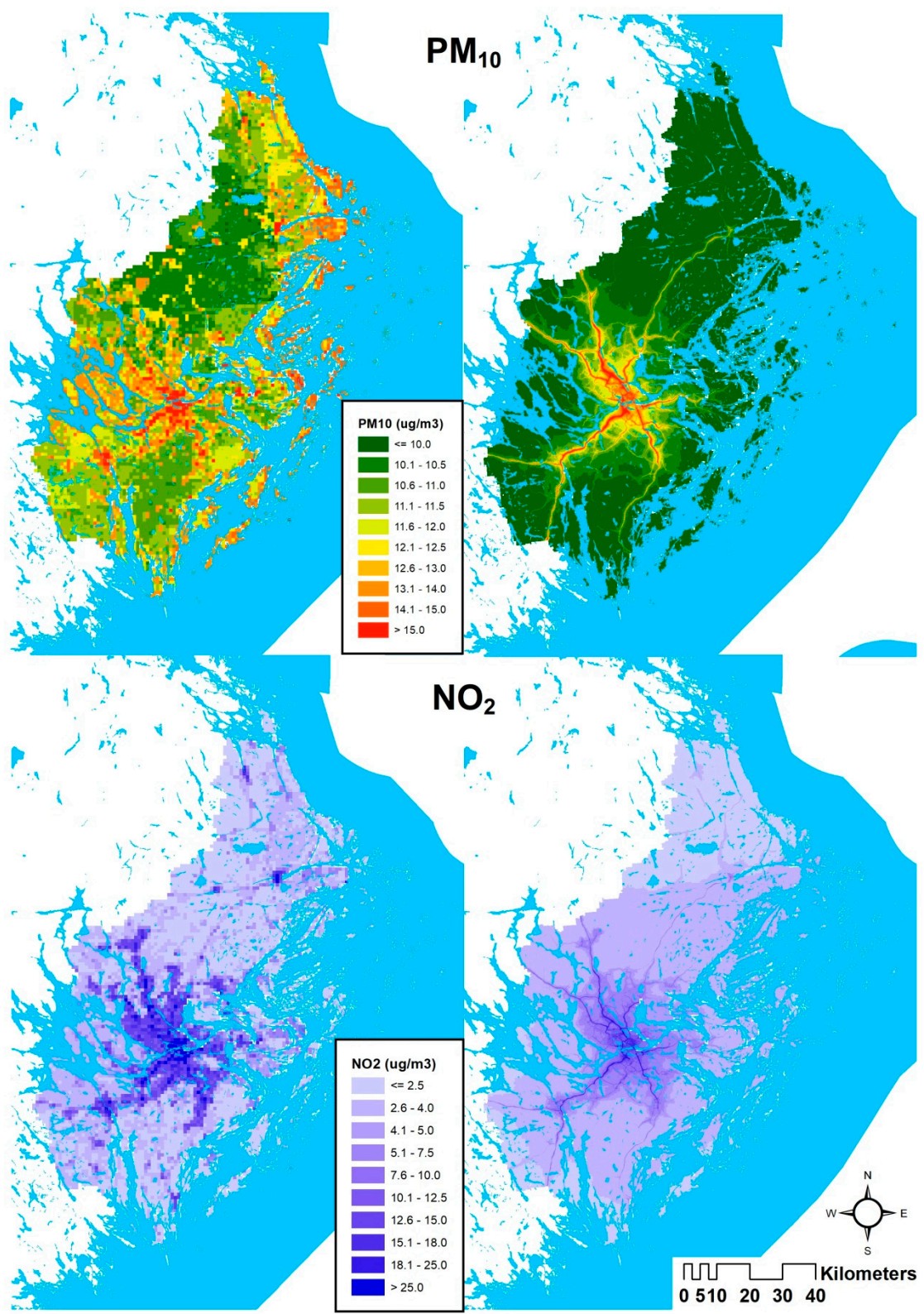

**Figure 5.** Prediction maps of the annual average concentrations of $PM_{10}$ (**top**) and $NO_2$ (**Bottom**) from random forest (**left**) and local dispersion (**right**) models in Stockholm County, year 2015.

### 3.7. Comparison with Previous Studies

In the last 15 years, there has been a proliferation of studies in which are developed spatiotemporal models to predict $PM_{10}$ and $PM_{2.5}$ daily concentrations over large geographical domains. The first ones to use columnar AOD to predict ground level particle concentrations applied simple approaches such as multivariate regression or correlational analyses [39–41]. Later, Kloog et al. proposed a mixed model framework aimed at capturing the temporally varying relationship between AOD and PM due to meteorological conditions in the USA [42,43]. The same methodology has been applied elsewhere [24,44,45]. More recently, machine-learning methods, such as random forests [20], gradient boosting [46], and neural network [47], have been developed due to their flexibility in handling nonlinear and interactive relationships among predictors and PM. This is a highly valued characteristic in situations where the joint relationship between daily particulate matter and multiple spatial and spatiotemporal predictors is only marginally understood. In the last years, outputs from dispersion models have been added to the list of potential predictors, and "ensemble" approaches have been proposed, under the assumption that the average of multiple base learners would benefit from the relative advantages of each one of them [48,49].

It is very difficult to compare the performance of so many different methods used in previous studies with the one proposed here. In summary, machine-learning methods seemed to outperform regression-based approaches, and ensemble designs only marginally improved model fit compared with individual base learners [48,49]. In this regard, we expect that the random forest methodology proposed here is the preferable option and a strength point of our study. On the other hand, the performance of the stage 3 training models was suboptimal in some cases (especially in held-out monitors), possibly because of a limited number of monitoring stations or the lack of key predictors such as national traffic data, emission data from industrial sources, etc. Despite this, the present models for $PM_{10}$ and $PM_{2.5}$ performed well in the main urban areas, where a large fraction of the population lives, and are therefore a valuable tool for investigating long-term (e.g., annual) and short-term (e.g., daily) health effects in these populations.

In contrast with $PM_{2.5}$ and $PM_{10}$, there are very few studies applying machine-learning methods to predict coarse PM [20], $NO_2$ [50], or $O_3$ [19] at fine spatiotemporal resolution over large geographical areas, and none of them conducted in Sweden. In a previous study conducted in Italy, we applied the same methodology proposed here to predict coarse PM, and we were able to predict 77% of $PM_{2.5-10}$ variability in OOB samples and 62% in held-out monitors [20]. De Hoogh et al. recently applied a similar approach to estimate $NO_2$ in Switzerland for the period 2005–2016 [50]. Their model explained ~58% ($R^2$ range, 0.56–0.64) of the variation in measured $NO_2$ concentrations, a value consistent with our OOB (74%) and held-out monitor (37%) CV-$R^2$. Di et al. developed a hybrid neural network methodology to predict daily ozone concentrations over the continental US and were able to predict 76% of $O_3$ variability, similar to our model in OOB samples (77%) [19].

Air quality dispersion modelling has been applied to quantify local and regional exposure to $PM_1$ and $PM_{10}$ in Sweden [51]. It was shown that long-range transport dominates average Swedish residential $PM_1$ and $PM_{10}$ levels, but for urban populations the contributions from urban and local traffic sources may dominate for residences close to heavily trafficked roads. The decreasing south to north and east to west concentration gradients of $PM_{10}$ across Sweden is very similar to the gradients obtained in the present study. Segersson et al. [33] modelled $PM_{10}$, $PM_{2.5}$, and black carbon (BC) in three urban areas in Sweden (Stockholm, Gothenburg, and Umeå) using Gaussian air quality dispersion models at a resolution of $100 \times 100$ m. The European, non-local contributions were taken from rural monitoring stations outside the cities or determined indirectly. Comparison between modelled and measured $PM_{10}$ concentrations at traffic and urban sites showed relative differences between annual averages between +11% to −16% (>0 means model overestimate). Corresponding values for $PM_{2.5}$ and BC were +24% to −49% and +13% to +14%, respectively. Korek et al. [52] applied a hybrid air pollution dispersion and land-use regression model (DM–LUR) using 93 biweekly observations of

$NO_x$ at 31 sites in the greater Stockholm area. The model predicted $NO_x$ concentrations ($R^2 = 0.89$) better than the DM without land-use covariates ($R^2 = 0.68$, P-interaction $< 0.001$).

*3.8. Strengths and Limitations*

The present study presents some important improvements compared with methods used in previous studies. First, it is the first study estimating daily concentrations of multiple air pollutants for the whole of Sweden (and one of the few estimating coarse PM, $NO_2$, and $O_3$ worldwide). Second, it applied a machine-learning methodology, the random forest, which proved to be highly efficient in other countries, often outperforming alternative methods [49]. Third, it combined multiple data sources among the predictors, including satellite-based parameters (AOD, NDVI, LAN) and atmospheric composition data from ensemble models. The main limitations to be acknowledged are the small numbers of monitoring stations (especially for $PM_{2.5}$ and $O_3$), the large fraction of missing AOD data (which had, however, a limited impact on the model as AOD was marginally important), and the weakness of some of the training models in predicting air pollution variability especially in held-out monitors, possibly due to the few monitors available, the little variability in observed concentrations, and the lack of key spatial predictors. Another limitation to mention is the high collinearity among several covariates added as predictors to the random forest model. However, while random forests are quite efficient in dealing with interactions, we further tried to reduce this problem by selecting only the subset of predictors which explained a non-negligible amount of variability in air pollutant concentrations. This resulted in CV estimates with little bias and no clear overfit of the data.

## 4. Conclusions

In this study we applied a multi-stage random forest methodology to predict daily concentrations of $PM_{10}$, $PM_{2.5}$, $PM_{2.5-10}$, $NO_2$, and $O_3$ for each squared kilometer of Sweden over the period 2005–2016. We combined satellite data, atmospheric composition variables, land-use terms, meteorological parameters, and population density as predictors of air pollution variability over space and time. Our models displayed negligible bias and were able to predict most of the variability, with cross-validated $R^2$ in the range of 0.64–0.77 for out-of-bag samples and 0.37–0.60 for held-out monitors. While we believe that our models' outputs should never replace measurements from operating monitoring networks, the estimates of spatial (e.g., annual means) and temporal (e.g., daily means) variability of multiple air pollutants as those provided here will allow the design of future epidemiological studies in Sweden aimed at investigating both short-term and long-term health effects of air pollution not only in the major cites but also in suburban and rural areas, previously neglected in epidemiological investigations.

**Supplementary Materials:** The following are available online at http://www.mdpi.com/2073-4433/11/3/239/s1. Table S1: Results of the Stage 1 model relating PM2.5 and PM2.5-10 to co-located PM10: descriptive statistics and cross-validation fit; Table S2: Results of the Stage 2 model relating MAIAC AOD to co-located CAMS AOD: descriptive statistics and statistics of model fit in OOB predictions; Table S3: Results of the Stage 3 model: statistics of model fit in predictions from "Out-of-bag" (OOB) samples; Table S4: Results of the Stage 3 model: statistics of model fit in predictions from 10-fold cross-validation by monitors; Table S5: Distribution of $PM_{10}$ and $NO_2$ concentrations from Random Forest (RF) and Dispersion Model (DM), Stockholm, 2015.

**Author Contributions:** Conceptualization, M.S. and T.B.; methodology, M.S.; software, M.S. and M.R.; validation, C.J. and P.G.; formal analysis, M.S.; investigation, M.S.; resources, M.D. and P.M.; data curation, M.S., A.S. and K.d.H.; writing—original draft preparation, M.S.; writing—review and editing, C.J., P.G., I.K. and T.B.; visualization, M.S.; supervision, T.B.; project administration, M.D. and T.B.; funding acquisition, M.S.. All authors have read and agreed to the published version of the manuscript.

**Funding:** This research received no external funding.

**Conflicts of Interest:** The authors declare no conflict of interest.

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
