# Peer review of "A Random Forest Approach to Estimate Daily Particulate Matter, Nitrogen Dioxide, and Ozone at Fine Spatial Resolution in Sweden"

_atmosphere, doi:10.3390/atmos11030239_

Round 1
Reviewer 1 Report
The authors attempt to model all available data for Sweden from 2005 to 2016 to individually describe air pollutants particulate matter, nitrogen dioxide and ozone for each square kilometer of land area. The data contains spatiotemporal predictor variables - Aerosol Optical Depth (AOD), meteorological data, atmospheric data and NDVI index. Also as spatial predictors are used resident population, ISA, LAN, land cover, road density and elevation. Additionally (though clearly unspecified) time data such as month, days of the week (unclear why of the week?) and others are used, as well as lagged predictors for the considered pollutants. Nonparametric Spearman's rho correlation and machine learning technique random forest (RF) are applied. Most likely, RF regression is applied, as it is mentioned that relationships are sought and regression curves are given. It should be made clear what and why was used from the results of the quoted R package ,ranger’ used, i.e. from ensembles of classification, regression, survival and probability prediction trees.
In general, many individual models were obtained, including separately by year, by season, and so on. General maps of Sweden are presented to illustrate obtained model pollution levels. In fact, Sweden is one of the EU member states that has no problem with major air pollution, according to European Commission reports (https://www.eea.europa.eu/publications/air-quality-in-europe-2017).
I consider it appropriate to make the following comments:
Table 2 shows that the importance rank of the last 11 predictors (major roads lengths and below this - have low ranks (> = 10), except for the % urban area for O3. This means that they are not significant and most likely impair the RF models. The same goes for the predictors: dew point temperature and snow albedo. Results for RF models without these low significant predictors are missing. This calls into question the use of many variables, most of which are statistically negligible. Also the 91-96% missing values for AOD is not acceptable. The R squared in Table S3 are not impressive. Rather, the RF models have not been well built. Again in Table 2, the Spearman rho correlation coefficients between PM10 and other pollutants with the predictors are extremely small in value and very strange. For example, that there is no correlation between PM10 and air temperature, or between PM10 and wind direction, etc. There is nothing mentioned about the p-value of these coefficients. The use of lagged variables with three backward moments is not justified; there is no study of possible autocorrelation. But especially for PM10, PM2.5 three lagged variables could not be explained easily by the chemical reactions in the air. Excessive collinearity between variables is not recommended for any type of regression, including tree-based methods (RF, CART, etc.). Evidence is not provided to show the stability of the presented models. In essence, section 3 for Results “reads” the table results, with a number of trivial conclusions. For instance, O3 is well known to has the opposite trend since it is a secondary pollutant for which PM10, NO2, PM2.5 are precursors. This is omitted. In Table S5, the large difference between Spearman's rho (RF, DM) = 0.22 for PM10 and rho (RF, DM) = 0.75 for NO2 is not explained. In general, it is not clear what is new about the presented approach and the resulting models compared to the cited similar models from [20, 24] and others by some of the same authors. Except the data used for Sweden. How are the maps in Figures 3 and 5 different or more informative from existing measurement maps, for example on the WHO website? What is the possible application of the obtained models? Apparently the authors do not give any idea.It gives the general impression that RF is applied at the level of information technology without thorough knowledge and mastering of RF and without a clear vision about the application of the models.
I greatly hesitate in my opinion about this paper. It's more neutral to negative.
Reviewer 2 Report
Atmosphere-672268
Title: A Random Forest approach to estimate daily particulate matter, nitrogen dioxide and ozone at fine spatial resolution in Sweden
This study described identifying that the results obtained provide new and compelling evidence on the short-term and long-term effects of air pollution on human health at the national level in Sweden. The study framework is not clear to me. Why only these factors are chosen for the study purpose, and why the other factors related to the short-term and long-term effects of air pollution on human health factors associated with air pollution perspectives are not included are not discussed. The authors should include the reasons for not including the other factors as the effects of environmental threats from a socioecological approach, as part of the study's limitations. Please see the following references: Kim, J., Kim, G. & Choi, Y. Effects of air pollution on children from a socioecological perspective. BMC Pediatr 19, 442 (2019) .

Reviewer 3 Report
There is no reason to suppose that health effects woulid be any different in rural areas vs cities. Instead, the dose may be different but not the health effects per se with regard to NO2. Perhaps another weakness in this study is that it cannot really substitute for actual air sampling results
end
Round 2
Reviewer 1 Report
I read the author's answers and I could refuse to review the revised paper. In my opinion the analyses are still superficial, models are unnecessarily overloaded and complicated, the analysis for autocorrelation are wrong.